Gut microbiota of obese and diabetic Thai subjects and interplay with dietary habits and blood profiles

Gruneck Lucsame 1 2
Kullawong Niwed 2 3
Kespechara Kongkiat 4
Popluechai Siam 1 2 siam@mfu.ac.th
1 School of Science, Mae Fah Luang University , Muang, Chiang Rai , Thailand
2 Gut Microbiome Research Group, Mae Fah Luang University , Muang, Chiang Rai , Thailand
3 School of Health Science, Mae Fah Luang University , Muang, Chiang Rai , Thailand
4 Sooksatharana (Social Enterprise) Co., Ltd. , Muang, Phuket , Thailand
Landa Blanca
Electronic publication date: 2020 Aug 3
Publication date: 2020
Volume: 8
Electronic Location ID: e9622
Received 2020 Mar 12; Accepted 2020 Jul 7
Copyright: © 2020 Gruneck et al.
Copyright year: 2020
Copyright holder: Gruneck et al.
License: This is an open access article distributed under the terms of the Creative Commons Attribution License, which permits unrestricted use, distribution, reproduction and adaptation in any medium and for any purpose provided that it is properly attributed. For attribution, the original author(s), title, publication source (PeerJ) and either DOI or URL of the article must be cited.
License URL: https://creativecommons.org/licenses/by/4.0/

Keywords: BMI, Fecal gut microbiota, Obesity, Next-generation sequencing, Thailand, Type 2 diabetes mellitus

Funding: Mae Fah Luang University This work was financially supported by Mae Fah Luang University via the Gut Microbiome research group. The funders had no role in study design, data collection and analysis, decision to publish, or preparation of the manuscript.

==============================
Obesity and type 2 diabetes mellitus (T2DM) have become major public health issues globally. Recent research indicates that intestinal microbiota play roles in metabolic disorders. Though there are numerous studies focusing on gut microbiota of health and obesity states, those are primarily focused on Western countries. Comparatively, only a few investigations exist on gut microbiota of people from Asian countries. In this study, the fecal microbiota of 30 adult volunteers living in Chiang Rai Province, Thailand were examined using next-generation sequencing (NGS) in association with blood profiles and dietary habits. Subjects were categorized by body mass index (BMI) and health status as follows; lean (L) = 8, overweight (OV) = 8, obese (OB) = 7 and diagnosed T2DM = 7. Members of T2DM group showed differences in dietary consumption and fasting glucose level compared to BMI groups. A low level of high-density cholesterol (HDL) was observed in the OB group. Principal coordinate analysis (PCoA) revealed that microbial communities of T2DM subjects were clearly distinct from those of OB. An analogous pattern was additionally illustrated by multiple factor analysis (MFA) based on dietary habits, blood profiles, and fecal gut microbiota in BMI and T2DM groups. In all four groups, Bacteroidetes and Firmicutes were the predominant phyla. Abundance of Faecalibacterium prausnitzii, a butyrate-producing bacterium, was significantly higher in OB than that in other groups. This study is the first to examine the gut microbiota of adult Thais in association with dietary intake and blood profiles and will provide the platform for future investigations.

Introduction

Over the past decades, obesity has been recognized as a global epidemic that threatens quality and length of life. Obesity has also been recognized as a risk factor for developing non-communicable diseases (NCDs) including cardiovascular diseases, cancers, diabetes and lung diseases (Caballero, 2007; Misra & Khurana, 2011; Webber et al., 2012). The prevalence of obesity is increasing globally in populations living in both developed and developing countries. However, the trends in obesity vary across continents with respect to economic progress (from high- to low-income countries) (Jaacks et al., 2019). Inequitable income reflecting an individual’s socioeconomic status seems to be one of the major factors that drive disparities in lifestyle-related health, particularly dietary behavior. In high-income countries (e.g., Western Europe and the United States), fresh foods are less affordable compared to processed foods among people with lower wages, which is a contributing factor to the increased prevalence of obesity (Drewnowski, 2009; Harrison & Taren, 2018). The inverse correlation pattern between obesity and rate of income have been reported repeatedly in low-income and middle-income countries (Wang, 2001; Swinburn et al., 2011; Pampel, Denney & Krueger, 2012; Harrison & Taren, 2018). A dietary habit has been shifted among middle-income countries at a brisk pace. Traditional diet is rapidly being replaced by consumption of processed foods in parallel with increasing urbanization. Exposure to high-fat diets in relation to socioeconomic status revealed that urban children tended to become more obese than children living in rural environments (Kisuse et al., 2018), an observation matched in urban workers (Xanthos, 2015). These patterns of dietary transition regarding convergence to obesogenic diets, which are energy-dense lead to obesity-related complications. The association of urbanization with obesity may thereby increase the burden of NCDs among populations (Low, 2016).

Body mass index (BMI) is widely used to assess health status based on weight and height. In general, higher BMI (overweight and/or obesity) is an indicator of increased risk of developing a range of conditions, including metabolic disorders, such as type 2 diabetes mellitus (T2DM) (Bays, Chapman & Grandy, 2007; Han & Boyko, 2018). Weight fluctuation is commonly associated with dietary consumption, and increasing food intake overtime, can lead to gaining weight. Westernized diets, which have high content of fats, sugars and sodium, but are deficient in fiber, increase risk of obesity along with its comorbidities including T2DM, heart disease and cancer (Mozaffarian et al., 2011; Manzel et al., 2014; Kopp, 2019). A strong relationship between BMI and T2DM has also been reported; the chance of developing T2DM increases in parallel with increasing BMI (Bays, Chapman & Grandy, 2007; Ganz et al., 2014; Al-Goblan, Al-Alfi & Khan, 2014; Gray et al., 2015). Currently, the interplay between BMI and microbes is a primary focus of research in determining host health traits.

The human gut harbors a large population of microorganisms, the gut microbiota, which exert a notable influence on the host in modulating energy balance (host metabolism and energy uptake). Numerous studies have suggested that bacteria residing within the human digestive tract are associated with health and disease states, and accordingly, they are involved in various host functions such as metabolism and immune system (Macpherson & Harris, 2004; Sekirov et al., 2010; Tremaroli & Bäckhed, 2012; Nicholson et al., 2012; Bull & Plummer, 2014; Leung et al., 2016; Tang & Hazen, 2016). Despite being abundant, an imbalance or disruption of the human gut flora can have a significant impact on disease susceptibility or occurrence (Manichanh et al., 2006; Clemente et al., 2012; Carding et al., 2015; Zhang et al., 2015b; Belizário & Faintuch, 2018). Among the commonly and consistently reported findings, the genera of Bifidobacterium, Bacteroides, Faecalibacterium, Akkermansia and Roseburia have been negatively associated with T2DM, while the genera of Ruminococcus, Fusobacterium, and Blautia were positively associated with T2DM (Gurung et al., 2020). Several studies have also demonstrated the crucial role of dysbiosis of intestinal microbiota in correlation with NCDs. Recent evidence suggests that alteration of the gut microbial composition may predispose the host to obesity and diabetes (Bäckhed et al., 2005; Hur & Lee, 2015; Serino et al., 2017). Evidence from mice and human studies have suggested that dysbiosis increases energy extraction from diet and enhances host energy harvest. Dysbiosis also induces obesity-associated inflammation in the host. The complexity of bacterial communities is significantly reduced in obese individuals (Le Chatelier et al., 2013). Restoring the lost complexity has been found to reduce metabolic disorders in animals (Yin et al., 2010; Gauffin Cano et al., 2012; Everard et al., 2013; Bubnov et al., 2017), and diets are considered as one of the major factors that contribute to the gut microbial community (Rinninella et al., 2019).

Diet is one of the most prominent external factors that does not only affect composition and abundance of gut microbiota, but also overall health. A relationship between diet and gut microbiota composition has been previously documented, whereby changes in gut microbial communities are influenced by variations in dietary components (Flint et al., 2012; David et al., 2014; Xu & Knight, 2015). Thus, the response of microbiome to diet potentially contributes to health status (Riaz Rajoka et al., 2017; Hughes et al., 2019). Thailand is used to be seen as a “lean nation”. During the last decade, the diet of Thai people has been changing and becoming more westernized due to economic development upgraded to a developing upper-middle-income country (World Bank, 2011), higher income and globalization. These foods have a high calorie but low nutritional content and their excessive consumption has been linked to obesity. The contemporary prevalence of overweight and obesity in Thai adults is 40.9% (Jitnarin et al., 2011). Moreover, NCDs cause 71% of total deaths in Thailand (WHO, 2011). Although the microbiota patterns of obese adult Asian populations from some Asian countries such as China, India and Japan (Kasai et al., 2015; Zhang et al., 2015a; Ahmad et al., 2019) are available, similar information on Thai gut microbiota is still limited. Given the unique culture and gastronomical lifestyle of Thailand, the present study aimed to (1) establish gut microbiota baselines of lean, overweight, obese, and T2DM in Thai populations and (2) explore associations of specific components of the gut microbiota with dietary habits and blood profiles in these populations.

Materials and Methods

Ethics statement

This study was approved by the Ethics Committee of Mae Fah Luang University (Ethics license: REH60075). The subjects were informed about the scope of the research project a day before participation using Thai-version information sheets. Written informed consents were obtained from all participants before sample and questionnaire collections.

Study subjects

The study included 30 subjects from Chiang Rai province located in Northern Thailand, which were considered to be representative of Thai population. Subject recruitment was conducted by voluntary participation through community clinics in the province and was carried out in July 2017. Subjects were divided into two major groups including diabetics (T2DM) and non-diabetics. Of these, seven subjects were placed in the T2DM group based on their morning fasting blood sugar level (cutoff level of > 126 mg/dL) (Reinauer et al., 2002) and irrespective of BMI. Twenty-three (non-diabetic) subjects were classified according to BMI using the criteria set by the World Health Organization Western Pacific Region (WHO, 2004), as follows: underweight (BMI < 18.5), normal or lean (18.5 ≤ BMI < 24.9), overweight (25.0 ≤ BMI < 29.9), and obese (BMI ≥ 30). Voluntary samples were taken from each group to meet a quota as follows: seven for T2DM (female (n = 6), male (n = 1)); eight for lean (female (n = 6), male (n = 1)), for overweight 8 (female (n = 6), male (n = 1)) for overweight, and seven for obese (female (n = 3), male (n = 4) and 4 men). Subjects that reported use of antibiotics (a duration of 6 months) and/or experienced diarrhea (a duration of 1 month) were excluded from the study. Average characteristics of the subjects that participated in this study are shown in Table 1. Statistical significance of each characteristic between groups (except gender) were assessed by One-way ANOVA test, followed by a post-hoc test for unequal sample size (Tukey–Kramer at a confidence interval of 0.95). The Fisher’s exact test was applied for a gender variable. The statistical analysis was performed using an R software package (stat) version 3.6.1 which Benjamini–Hochberg procedure was applied for multiple-test correction using multcomp package (version 1.4-10). Blood profiles, high-density lipoprotein (HDL) cholesterol and fasting glucose levels were also selected for further multivariate analysis.

Table 1 Characteristics of the subjects that participated in this study.

One-way ANOVA test (p < 0.05) was used to compute the difference of mean for each characteristic across groups. Superscript letters indicate statistical comparison between the means of groups at a confidence interval of 0.95.

Characteristic	Total (n = 30)	Lean (n = 8)	OV (n = 8)	OB (n = 7)	T2DM (n = 7)	p-valuei	
Age (years)	48.37 ± 13.36	49.13 ± 7.66	40.43 ± 9.69e	42.00 ± 17.82g	60.29 ± 4.83e,g	0.0299	
Gender (male/female)	7/23	1/7	1/7	4/3	1/6	0.2259j	
BMI (kg/m2)	27.33 ± 4.11	23.70 ± 1.30a,b	27.52 ± 1.15b,d	33.28 ± 2.15a,d,h	25.55 ± 3.25h	p < 0.000	
Weight (kg)	69.18 ± 15.06	58.75 ± 6.51a	67.36 ± 8.92d	90.86 ± 12.56a,h	62.14 ± 5.77h	p < 0.000	
Height (cm)	158.53 ± 9.10	157.25 ± 6.78	156.14 ± 9.22	164.86 ± 9.66	156.43 ± 8.45	0.233	
Blood pressure (BP)							
Systolic BP (mmHg)	138.23 ± 15.84	136.50 ± 16.78	135.29 ± 10.05	138.29 ± 19.25	139.71 ± 13.67	0.986	
Diastolic BP (mmHg)	87.50 ± 9.14	87.50 ± 9.75	89.71 ± 11.50	84.57 ± 7.72	87.00 ± 5.81	0.693	
Total cholesterol (mg/dL)	206.90 ± 38.21	214.50 ± 43.24	225.57 ± 40.44	194.57 ± 35.18	192.57 ± 21.74	0.366	
LDL cholesterol (mg/dL)	115.23 ± 36.39	112.63 ± 36.42	138.86 ± 37.67	114.00 ± 32.19	97.57 ± 28.36	0.296	
HDL cholesterol (mg/dL)	60.63 ± 17.08	75.63 ± 14.70f	60.14 ± 14.53	42.57 ± 6.37f	67.71 ± 12.89	p < 0.000	
Triglyceride (mg/dL)	154.37 ± 79.61	131.13 ± 71.71	132.14 ± 41.92	188.57 ± 100.74	165.71 ± 78.87	0.518	
Fasting glucose (mg/dL)	111.60 ± 34.60	96.13 ± 12.94c	97.00 ± 8.09e	110.14 ± 32.19	146.29 ± 50.37c,e	0.0142	
Note:

aLean < OB, bLean < OV, cLean < T2DM, dOV < OB, eOV < T2DM, fOB < Lean, gOB < T2DM, hT2DM < OB, ione-way ANOVA test, jFor gender, the statistical significance was assessed by post-hoc pairwise Fisher’s exact test. a,b,c,d,e,f,g,hStatistically significant differences were observed (Tukey–Kramer post-hoc test, q < 0.05).

Food frequency questionnaire

Dietary intake variables were collected using food frequency questionnaire (FFQ). The questionnaire contained 25 items of different food types including rice vermicelli, a traditional food of northern Thailand (La-ongkham et al., 2015). Records of frequency of consumption of yogurt/cheese/fermented milk, and fruits were missing for one subject (lean group). Frequencies were categorized into the following six levels: every day, 5–6 days a week, 3–4 days a week, 1–2 days a week, less than once a week, and never. The statistical significance of differences in the mean ranks among groups was determined using Kruskal–Wallis rank sum test with post-hoc analysis (Dunn’s test of multiple comparisons, p-value adjusted with the Benjamini–Hochberg method (hereafter referred to as q-value)). The frequencies of dietary consumption of each group are summarized in Table S1.

Fecal sample collection and DNA extraction

Fecal samples of all volunteers were collected in a sterilized container and immediately stored at −20 °C until further use. Total genomic DNA from fecal samples was extracted using the innuPREP Stool DNA Kit (Analytik Jena Biometra, Jena, Germany) following the manufacturer’s guidelines. Concentration and purity of DNA were evaluated on 1% agarose gels. Spectrophotometry was applied to determine the DNA concentration (ng/µl) by the Take 3 Micro-Volume Plate (Biotek, Winooski, VT, USA). Total DNA per gram of fecal wet weight was calculated and recorded.

Amplicon generation, library preparation and sequencing

The hypervariable region V3–V4 of the 16S rRNA gene was amplified using specific primers (16S V3–V4: 341F: 5′-CCTAYGGGRBGCASCAG-3′, 806R: 5′-GGACTACNNGGGTATCTAAT-3′) (Klindworth et al., 2013) with the barcode. All PCR reactions were carried out using Phusion® High-Fidelity PCR Master Mix (New England Biolabs, Ipswich, MA, USA). PCR products were run using electrophoresis on a 2% agarose gel for detection. Samples that showed a band between 400 and 450 bp were chosen for further experiments. PCR products were mixed in equidensity ratios. Then, mixture PCR products were purified with Qiagen Gel Extraction Kit (Qiagen, Germany). Sequencing libraries were generated using NEBNext ® Ultra DNA Library Pre Kit for Illumina, following manufacturer’s recommendations and index codes were added. Library quality was assessed using Qubit@ 2.0 Fluorometer (Thermo Scientific, Waltham, MA, USA) and Agilent Bioanalyzer 2100 system. Libraries were sequenced on the Miseq platform (Illumina, San Diego, CA, USA) at Novogene (Beijing, China) during September 2017 and 250 bp paired-end reads were generated. Further downstream steps included data analysis using Qiime (version 1.7.0), OTU clustering and taxa annotation, alpha and beta diversity analysis.

Data analysis

Paired-end reads were assigned to samples based on their unique barcode and truncated by cutting off the barcode and primer sequence. Paired-end reads were merged using FLASH (V1.2.7, http://ccb.jhu.edu/software/FLASH/) (Magoč & Salzberg, 2011). Splicing sequences were called raw tags. Quality filtering on the raw tags was performed under specific filtering conditions to obtain high-quality clean tags (Bokulich et al., 2013) according to the QIIME (version 1.7.0, http://qiime.org/index.html) (Caporaso et al., 2012) quality-controlled process. Tags were compared with the reference database (Gold database, http://drive5.com/uchime/uchime_download.html) using UCHIME algorithm (UCHIME Algorithm, http://www.drive5.com/usearch/manual/uchime_algo.html) (Edgar et al., 2011) to detect chimera sequences, all of which were removed (Haas et al., 2011). The raw sequence data is available at NCBI SRA with BioProject accession number PRJNA610672 (BioSample accession numbers SAMN14309526–SAMN14309555).

OTU cluster and species annotation

Sequence analysis was performed using Uparse software (Uparse version 1.0.1001, http://drive5.com/uparse/) (Edgar, 2013). Sequences with ≥97% similarity were assigned to the same OTU. A representative sequence for each OTU was screened for further annotation. Sequences were queried against the Greengenes Database version gg.13.5 (http://greengenes.lbl.gov/) (DeSantis et al., 2006; Wang et al., 2007) to obtain taxonomic information. Newly generated OTUs were aligned using MUSCLE software (Version 3.8.31, http://www.drive5.com/muscle/) (Edgar, 2004) and phylogenetic trees were generated. The OTU annotation tree was visualized using a custom R package (developed by Novogene Co., Ltd., Beijing, China). OTU abundance information was obtained by normalizing the sequence number corresponding to the sample with the least sequences (OTU counts rarefied to 103,744 reads per sample). Subsequent analysis of alpha diversity and beta diversity were all performed basing on this output normalized data. The relative abundance of gut bacteria between sample groups was compared by the unpaired two-samples Wilcoxon test and multiple comparisons were adjusted with Benjamini–Hochberg method (q < 0.05) using R software package (stats) version 3.6.1.

Alpha diversity

Alpha diversity was applied to analyze complexity of species diversity for each sample using the following indices: ACE, Chao1, observed-species, Shannon, Simpson and Good’s coverage. The unpaired two-samples Wilcoxon test with Benjamini–Hochberg procedure (q < 0.05) was used to compare alpha diversity indices between groups for statistical differences. Comparisons were visualized as a box plot by ggplot2 (Wickham, 2009) in R software (version 3.6.1).

Beta diversity analysis

Beta diversity analysis was used to evaluate differences of fecal samples in bacterial community structure between BMI and T2DM groups. Beta diversity was calculated using both weighted and unweighted unifrac. Principal Coordinate Analysis (PCoA) was performed to get principal coordinates and visualize from complex, multidimensional data. PCoA analysis was displayed by WGCNA package, stat package and ggplot2 package (Wickham, 2009) in R software (version 2.15.3). Multi-response permutation procedure (MRPP) was used to determine dissimilarities of microbial community structure between groups implemented in the R package vegan (version 2.5–6) (Mielke & Berry, 2001). Unweighted Pair-group Method with Arithmetic Means (UPGMA) Clustering was performed as a type of hierarchical clustering method to interpret the distance matrix using average linkage. The relative abundance of OTUs that most likely explain the differences between groups was evaluated by LEfSe (linear discriminant analysis (LDA) Effect Size) analysis (Segata et al., 2011).

Firmicutes/Bacteroidetes ratios

The non-parametric Wilcoxon rank–sum test was performed to compare Firmicutes/Bacteroidetes ratios between groups (L, OV, OB and T2DM) (p < 0.05). The comparisons were visualized as a boxplot by ggplot2 (Wickham, 2009).

Multivariate statistical analysis

In order to gain a deeper insight into the dietary consumption profile of individuals in the different groups, we used multiple factor analysis (MFA) in FactorMineR version 1.42 (R software, version 3.6.1) (Lê, Josse & Husson, 2008). MFA is beneficial for simplifying and structuring variables into groups on the components (PCA-based eigenvalues). The MFA of dietary consumption in different BMI and T2DM groups included 25 variables (frequency of intake) that belonged to 4 variable groups (protein (8 variables), carbohydrate (6 variables), fiber (5 variables), and beverage (6 variables)). For evaluating the association between dietary habits, blood profiles and fecal gut bacteria in BMI and T2DM groups, 37 variables were included in the MFA, which the analysis of multidimensional distance between subjects was based on 25 variables pattern of food consumption (frequency of intake), 2 variables described blood profiles (HDL cholesterol and fasting glucose levels), and 10 variables belonged to relative abundance of fecal gut microbiota at genus level. An integration of confounding factors (gender and age) with other concerned variables (blood profiles, dietary habits and fecal gut microbiota) also implemented in the MFA analysis. The age variable was categorized into six groups (20–29, 30–39, 40–49, 50–59, 60–69 and over 70) according to Png et al. (2016). Therefore, the variations were revealed by the influence of each variable on the principle components. Results of multivariate data analyses were extracted and visualized by Factoextra version 1.0.5. We used the mixOmics R package version 6.10.2 (Rohart et al., 2017) to determine associations between microbial communities, dietary consumption and blood profiles in the different groups. We applied sparse Partial Least Square (sPLS) analysis to explore relationships between these variables in each study group (L, OV, OB and T2DM). The sPLS “canonical mode” was used to specify the microbial OTU that most correlated with diets and/or blood profiles (Lê Cao et al., 2008, 2009). The high-dimensional data sets were visualized with clustered image maps (González et al., 2013).

Results

Dietary consumption in different BMI groups and T2DM subjects

Dietary habits of the different groups were determined to enquire whether the observed data could support the gut microbial profile. Based on the frequency of consumption records, none of the subjects in any of the groups differed in the intake of dietary protein, carbohydrate, fiber and beverage with some exceptions: significant differences were noted regarding consumption of chicken (OB-T2DM comparison, q < 0.01), rice vermicelli (L-T2DM comparison, q < 0.05) and fermented fruits or vegetables (L-OV and L-T2DM comparisons, p < 0.05 for both).

The results of MFA revealed that individuals with T2DM displayed a notable variation on frequency of food consumption from the rest of the groups. This was discernible on the factor map indicating the first two dimensions accounting for 31.6% of variance (Fig. S1). The ellipse of L, OV and OB groups had a strong overlap compared to the ellipse representing T2DM group with 95 % confidence. These results indicate a lower variability of dietary consumption among the different BMI groups when compared to the T2DM group (coordinate = −1.42, p < 0.001). The frequency of fermented fruit/vegetable consumption significantly described the first dimension (r = 0.60, p < 0.001), and that variation in food consumption among groups supported the reduction of this particular type of dietary fiber intake in subjects having type 2 diabetes.

Blood profiles in BMI and T2DM groups

Information regarding subject status of the study is shown in Table 1. There were statistical differences between the average age, BMI, weight, HDL cholesterol, and fasting glucose in four study groups assessed by Tukey–Kramer post-hoc test (q < 0.05). HDL cholesterol was significantly lower in the OB groups compared to L (q < 0.001), OV and T2DM groups (q < 0.05), and was also significantly lower in OV vs. L groups (q < 0.05). Unsurprisingly, fasting glucose level increased with increasing BMI with highest level in T2DM group (q < 0.05).

Composition of prokaryotic fecal microbiota in BMI and T2DM groups

A total of 3,408,383 reads were obtained from 16S rRNA amplicon sequencing with an average of 113,613 reads per sample for a total of 30 samples. Using 97% identity criterion for determining OTUs, we obtained 504.33 ± 33.15 OTUs per sample (range 454–579 OTUs). Gut microbiota of all samples was classified into 995 OTUs, 145 genera, 82 families, 53 orders, 31 classes and 18 phyla. Shared and unique OTUs among different groups are shown in Venn diagram (Fig. 1). The total number of OTUs presented in the L, OV, OB and T2DM groups were 832, 811, 753, 852, respectively. The number of shared OTUs in all groups was 588; 729 OTUs (22.44%) were shared between L and OV groups, 752 OTUs (23.15%) between L and T2DM groups, 671 (20.65%) between L and OB groups, 730 (22.48%) between OV and T2DM groups, 657 (20.22%) between OV and OB groups, and 629 (19.37%) between T2DM and OB groups (Fig. 1A). Regarding the number of non-shared OTUs, non-diabetic subjects (merging OTUs of L, OV and OB groups) had four times more than diabetic subjects (T2DM). Specifically, 139 non-shared OTUs (17%) were associated with the non-diabetic group, whereas 33 OTUs (4%) were uniquely present in T2DM subjects (Fig. 1B). The rarefaction curves of microbial diversity estimators for thirty samples reached plateau phase, indicating that most microbial species had been detected in all samples (Figs. S2A−S2C).

Figure 1 Venn diagram showing the number of microbial compositions according to OTU classification.

(A) Fecal microbiome OTUs in four groups (L, OV, OB and T2DM). (B) OTU distribution in non-diabetic (L, OV and OB) and diabetic subjects (T2DM).

Fecal microbiome community diversity (richness and evenness) in the four groups was characterized using ACE, Chao1, observe-species, Shannon, Simpson and Good’s coverage. Sequencing data and alpha diversity indices in each sample are presented in Tables S2 and S3. Significant differences of overall bacterial community structure across the four groups were found in the ACE, Chao1, and observe-species indices (Fig. 2). Specifically, microbial communities of L and T2DM groups had significantly greater species richness as compared to those in OB. No significant differences in the diversity of communities (species richness and evenness) were found across the four groups by Shannon and Simpson indices, suggesting a similar pattern of the community composition in all groups. Nevertheless, microbial communities of all four groups had high species-level diversity as indicated by Simpson index, the value of which approached 1. This implies that as species richness and evenness increased, diversity also increased.

Figure 2 Boxplots of alpha diversity indices in each group (L, OV, OB and T2DM).

Alpha diversity measured by (A) ACE index, (B) Chao1 index, (C) observed species, (D) Shannon index, and (E) Simpson index. The paired comparisons were determined using Wilcoxon rank–sum test adjusted for multiple testing with the Benjamini–Hochberg method (asterisks indicate q < 0.05).

The top ten phyla of microbial communities across the four groups were Bacteroidetes, Firmicutes, Proteobacteria, Fusobacteria, Actinobacteria, Verrucomicrobia, Cyanobacteria, Tenericutes, Elusimicrobia, and TM7. No significant differences were detected among groups (p < 0.05) (Figs. S3A and S3B). Top ten of bacterial genera with high relative abundance were used to construct phylogenetic relationships (Fig. S3C). Based on the similarity threshold, some bacterial species related to Prevotella genus were clustered in (Prevotella) as their discrete lineages distinct from other known species within this genus. The representative OTUs assembled in (Prevotella) consist of Prevotella tannerae (1 OTU), Uncultured bacterium (7 OTUs) and Prevotellamassilia timonensis (2 OTUs). The latter is a newly identified bacterial species in the human gut (Ndongo et al., 2016).

We further compared the median differences of the Firmicutes:Bacteroidetes ratios between groups with 95% confidence interval at phylum level (Fig. S3D). No statistically significant difference was noted among any of the groups. Among the top ten dominant genera, Prevotella and Bacteroides accounted for the largest proportion in all sample groups (Figs. 3A and 3B). Several significant differences among the dominant taxa were also found without the Benjamini–Hochberg method. Faecalibacterium showed significant differences in OV-OB (p = 0.044) and OB-T2DM (p = 0.011) comparisons (Fig. 3C), while (Prevotella) differed in L and OV comparison (p = 0.034) (Fig. 3D). Gut microbial alterations at species levels in four groups of samples (Figs. S4A and S4B) present significant differences of butyrate-producing bacteria (Faecalibacterium prausnitzii) in OV-OB (p = 0.044) and OB-T2DM (p = 0.011) comparisons (Fig. S4C), whereas there was a statistically significant difference between OV and T2DM group (p = 0.026) with respect to the relative abundance of Prevotella copri (Fig. S4D). Bacteroides coprophilus was also significantly different in OV-OB (p = 0.026) and OV-T2DM comparisons (p = 0.015). Analysis of statistical differences of microbial species abundance among groups by LEfSe showed marked differences between subject groups (Fig. 4). In the histogram, the colors represent taxa that were found to be more abundant in OB group compared to the other groups (both for the positive (Figs. 4A, 4C, and 4D) and the negative score (Fig. 4E)). In this regard, there were six taxa (Veillonellaceae, Prevotella, Dialister, Megamonas, Faecalibacterium, and Faecalibacterium prausnitzii), two genera (Clostridium and Prevotella) and one species (Bacteroides coprophilus) enriched in OB, T2DM and OV groups, respectively, while no enrichment of species was noted in the L group.

Figure 3 Gut microbial community abundance at the genus level.

(A) Bar plot of the relative abundance of top ten fecal gut microbiome at genus level presented in each subject. (B) Bar plot of the relative abundance of top ten fecal gut microbiome at genus level presented in each group. (C) Boxplot of relative abundance of Faecalibacterium across four groups. (D) Boxplot of relative abundance of [Prevotella] across four groups. Asterisks indicate p < 0.05, Wilcoxon rank–sum test without Benjamini–Hochberg method.

Figure 4 LEfSe analysis of fecal microbiota in BMI groups and T2DM group.

(A) LEfSe comparing microbiome between L, OV, and OB groups. (B) LEfSe comparing microbiome between L, OV, and T2DM groups. (C) LEfSe comparing microbiome between L, OB, and T2DM groups. (D) LEfSe comparing microbiome between OV, OB, and T2DM groups. (E) LEfSe comparing microbiome between L, OV, OB, and T2DM groups. Histogram of LDA scores showing taxa with significant differences among groups (LEfSe bar at species level, p < 0.05, LDA value > 4). Species whose LDA scores (the effect size) are larger than 4 were presented as bars in different colors (blue, red, and yellow).

Beta diversity analysis of fecal microbiota in BMI and T2DM groups

Qualitative (unweighted UniFrac) and quantitative beta (quantitative measure) diversity measures yielded substantially different perspectives on the factors (BMI groups and/or the disease) that may be involved in structuring bacterial diversity. Unweighted UniFrac showed less distance of samples than Weighted UniFrac (Fig. S5B). PCoA based on Unweighted UniFrac revealed clearer patterns of microbial variation (Fig. 5B). Gut microbial communities in OB group were significantly different from those of the T2DM group (p < 0.05), whereas no dissimilarity was observed in comparison with other groups. Furthermore, more similar community composition was observed in OB and OV groups with some overlaps also identified by UPGMA (Fig. S5B). Clustering analysis suggested association of BMI and/or a disease with the variations in bacterial community compositions among subjects. Conversely, PCoA and weighted UniFrac did not clearly discriminate microbial communities among groups implying that there was no strong association with BMI and/or T2DM in this population (Fig. 5A). When taking the relative abundance of each type of OTUs into account, the results displayed similarities in bacterial composition, suggesting that indistinguishable communities may result from the number of organisms collected in the dominant phylum (Firmicutes, Bacteroidetes and Proteobacteria) (Fig. S5A).

Figure 5 Beta diversity analysis of the OTUs at phylum level.

(A) PCoA based on Weighted UniFrac distance. (B) PCoA based on Unweighted UniFrac distance. Subjects from L, OV, OB and T2DM groups are labeled in red, green, black and blue color, respectively.

Associations between dietary habits, blood profiles, and fecal gut microbiota in BMI and T2DM groups

MFA revealed the variables that mostly contributed in explaining the variations regarding dietary habits, blood profiles, and the relative abundance of gut microbiota of subjects in different BMI groups as well as the T2DM group. The factor map of the MFA generated by data integration of all variables showed the significance of Dim 1 and 2 that explained 14.3% and 12.8 of observed variability, respectively (Fig. 6). The distinct or similar profile of individuals, indicated by the ellipses on both axes of the MFA, mainly resulted from the variation in the blood profiles: HDL cholesterol levels were lower in the OB group than those observed in the other groups (coordinate = −1.20, p < 0.05) and higher fasting glucose levels were firmly correlated with T2DM (coordinate = −1.54, p < 0.001) in comparison with the BMI groups. In addition to the blood profiles, the genera of bacteria that associated with Dim 1 included Fusobacterium (r = 0.55, p < 0.01), Bacteroides (r = 0.52, p < 0.01), Prevotella (r = −0.54, p < 0.01), and Faecalibacterium (r = −0.47, p < 0.01). Fusobacterium was associated with Dim 2 (r = 0.47, p < 0.01). For dietary consumption, dairy products and mixed rice variables were negatively correlated with Dim 1 (r = −0.63, p < 0.001) and Dim 2 (r = −0.64, p < 0.001), respectively. The second dimension was described by a beef variable with a correlation coefficient of 0.53 (p < 0.01). The relationships of the relative abundance of gut bacteria and the frequency of food intake were nevertheless interpreted as moderate correlation to both dimensions (moderate variance). Concerning the contribution of all variables to describe the differences between individuals, the current analysis suggested that blood profiles seemed to have most influence on the variability of 30 subjects with different BMIs or with T2DM. Furthermore, the association between dietary intake, blood profiles, and microbial OTUs at the genus level from sPLS analysis are additionally summarized in Table 2.

Figure 6 MFA analysis of dietary consumption, blood profiles, and fecal gut microbiota of subjects in different BMI groups and T2DM group.

The factor map presents the integration of dietary consumption, blood profiles (HDL cholesterol and fasting glucose level), and fecal gut microbiota (at genus level) of subjects in different BMI groups and T2DM group based on the MFA. The coordinates of the individuals are indicated by the 95% confidence ellipses including orange (L), blue (OV), green (OB), and purple (T2DM).

Table 2 sPLS analysis.

The association of taxonomic composition at the level of genus with dietary consumption and blood profiles in different BMI groups and T2DM group.

Bacterial phyla	Bacterial taxa (OTU at genus level)	Associated food groups/blood profiles	Association	Study group (s)	
Firmicutes	[Ruminococcus]	Pea/nut/bean	Negativea	OB	
Fish	Negativea	OB	
Chicken	Negativea	OB	
Brown rice	Positiveb	OV	
Green vegetable	Negativea	OB	
Fruits	Negativea	OB	
Fasting glucose	Negativec	OB	
HDL cholesterol	Negativec	T2DM	
Roseburia	Pork	Negativea	OV	
Chicken	Negativea	OV	
Brown rice	Positiveb	L, OV	
Specified vegetables	Negativea	L	
Specified vegetables	Positiveb	T2DM	
Grain	Positiveb	T2DM	
Carbonate soft drink	Negativea	OB	
Juice	Positivea	T2DM	
Fasting glucose	Positived	L, OV, T2DM	
HDL cholesterol	Negativec	T2DM	
Faecalibacterium	Pork	Negativea	T2DM	
Brown rice	Positiveb	L	
Specified vegetables	Negativea	L	
Grain	Negativea	OV	
Juice	Positiveb	T2DM	
Fasting glucose	Positived	L, OV, T2DM	
Oscillospira	HDL cholesterol	Negativec	OV	
HDL cholesterol	Positived	OB, T2DM	
Bacteroidetes	[Prevotella]	Pea/ nut/ bean	Positiveb	OB	
Beef	Positiveb	OB	
Sticky rice	Positiveb	OB	
Specified vegetables	Positiveb	L, OB	
Grain	Positiveb	OB	
Juice	Positiveb	OB	
Fasting glucose	Negativec	L, OV	
Fasting glucose	Positived	OB	
Prevotella	Pork	Negativea	T2DM	
Sticky rice	Positiveb	OB	
Bread	Positiveb	T2DM	
Grain	Negativea	L	
Fermented fruits/vegetable	Positiveb	T2DM	
		Green vegetable	Negativea	T2DM	
Coffee	Positiveb	OV	
Alcohol	Negativea	OV	
Carbonate soft drink	Positiveb	T2DM	
Fasting glucose	Positived	L, OB	
HDL cholesterol	Negativec	OV	
Bacteroides	Brown rice	Positiveb	OB	
Bread	Negativea	T2DM	
Grain	Positiveb	L	
Specified vegetables	Positiveb	L	
Fruits	Negativea	T2DM	
Fasting glucose	Negativec	L	
HDL cholesterol	Positived	OV, OB, T2DM	
Fusobacteria	Fusobacterium	Pork	Positiveb	L, T2DM	
Fish	Positiveb	L	
Dairy products	Positiveb	T2DM	
Beef	Positiveb	T2DM	
Rice vermicelli	Positiveb	T2DM	
Green vegetable	Negativea	OB	
Fruits	Negativea	OB	
Alcohol	Positiveb	L	
Coffee	Negativea	OV	
Carbonate soft drink	Positiveb	OB	
HDL cholesterol	Positived	L, OV	
HDL cholesterol	Negativec	OB, T2DM	
Fasting glucose	Negativec	OV	
Proteobacteria	Escherichia	Brown rice	Positiveb	OV	
Juice	Positiveb	T2DM	
Fasting glucose	Positived	OV, T2DM	
HDL cholesterol	Negativec	T2DM	
Sutterella	Coffee	Negativea	L	
Juice	Negativea	T2DM	
HDL cholesterol	Positived	L, T2DM	
HDL cholesterol	Negativec	OB	
Fasting glucose	Negativec	OV, T2DM	
Fasting glucose	Positived	OB	
Notes:

a Negative correlation: correlation coefficient < −0.7 for dietary consumption.

b Positive correlation: correlation coefficient > 0.7 for dietary consumption.

c Negative correlation: correlation coefficient < −0.5 for blood profiles.

d Positive correlation: correlation coefficient > 0.5 for blood profiles.

Gender and age highly contributed to the variations of subjects in different BMI groups and T2DM group (Fig. S6). Integration of five variables (gender, age, blood profiles, dietary consumption, and the relative abundance of gut microbiota) by the MFA revealed that gender and age were the top two variables that highly explained individual variation in terms of blood profiles, dietary consumption, and the relative abundance of gut microbiota. The opposed pattern of T2DM females (coordinate = −3.12, p < 0.001) at age levels of four (50–59) and five (60–69) to Dim 1 was explained by the consumption of fish (r = −0.82, p < 0.001), whereas the consumption of fermented fruits or vegetables (r = 0.52, p < 0.01) and dairy products (r = 0.51, p < 0.01) as well as the abundance of Faecalibacterium (r = 0.49, p < 0.01) tended to be prevalent in OB females (coordinate = 1.92, p < 0.05). HDL cholesterol levels were negatively correlated to Dim 2 (−0.78), particularly among L females (coordinate = −3.0, p < 0.001) at the age level of three (40–49) (coordinate = −2.58, p < 0.001) that displayed higher HDL profile than OB and T2DM. In Dim 3, Escherichia was moderately correlated to the dimension (r = 0.51, p < 0.01) and was predominant in OB male at the age of 71 (coordinate = 3.25, p < 0.01). The distinct pattern of T2DM females marked in Dim 4 (coordinate = 1.52, p < 0.05) resulted from high fasting glucose levels with the correlation dimension of (r = 0.51, p < 0.01).

When all blood tests were taken into account, several important variables were maintained (Fig. S7), however, their variations were described by different dimensions as compared to the MFA in Fig. S6. Different profiles between OB females and T2DM females were illustrated in Dim 1, where the consumption of chicken (r = 0.70, p < 0.001), dairy products (r = 0.59, p < 0.001), and the abundance of Faecalibacterium (r = 0.53, p < 0.01) were less prevalent among T2DM females (coordinate = −2.99, p < 0.001) in comparison with OB females (coordinate = 2.21, p < 0.01). A distinct cluster of L females observed in Dim 2 (coordinate = −2.65, p < 0.001) was mostly described by HDL cholesterol levels (r = −0.72, p < 0.001) as being analogous to Dim 2 of Fig. S6. A moderate correlation of Escherichia dimension (r = 0.42, p < 0.05) also displayed in Dim 2 in parallel with OB male at the age of 71 (coordinate = 3.22, p < 0.01). Faecalibacterium (r = 0.44, p < 0.05) and FBS (r = 0.37, p < 0.05), however consistently presented in Dim 3, was predominant in OB females (coordinate = 1.90, p < 0.05). Interestingly, most OV individuals were negatively correlated with Dim 2 as resulted from LDL cholesterol levels (r = −0.45, p < 0.05) that tended to be higher in both OV male (coordinate = −2.92, p < 0.05) and females (coordinate = −1.51, p < 0.001), especially at the age level of two (30–39) (coordinate = −2.10, p < 0.001).

Exclusion of gender and age variables showed a strong overlap of non-diabetic (BMIs) and diabetic (T2DM) subjects, though no specific group was defined in relation to the variations of blood profiles and gut microbiota, indicating dispersal of variables across BMI and T2DM groups (Fig. S8). In Dim 1, the distribution of subjects on the MFA map was mainly influenced by the abundance of three major genera including Bacteroides (r = 0.68, p < 0.001), Prevotella (r = −0.65, p < 0.001), and Faecalibacterium (r = −0.65, p < 0.001). Cholesterol (r = 0.49, p < 0.01), triglyceride (r = 0.43, p < 0.05), and diastolic blood pressure (r = 0.43, p < 0.05) levels moderately correlated with the dimension. Furthermore, systolic blood pressure (r = 0.68, p < 0.001) and HDL cholesterol (r = −0.54, p < 0.01) levels mainly described the individual variance in the second dimension.

Discussion

Our study provides the first evaluation of bacterial gut microbiota composition in adult Thai subjects of various BMI and T2DM. Microbial diversity across four groups was examined using six indices based on richness and evenness. The BMI between OB-L and OB-T2DM, OB group was associated with a significant decrease in bacterial diversity across three indices (ACE, Chao1 and observed species), whereas a change of diversity was maintained in L and T2DM groups. This is in agreement with the previous study of Chinese subjects with different glucose intolerance statuses (normal glucose tolerance, prediabetes, T2DM) (Zhang et al., 2013). The observed reduced-bacterial diversity in OB groups is consistent with previous findings, in that obese subjects exhibited lower alpha diversity, when compared with non-obese subjects (Turnbaugh et al., 2009; Le Chatelier et al., 2013). Accordingly, obese individuals in a Korean population displayed lower gut bacterial diversity (phylogenetic diversity index) than normal weight and overweight individuals (Yun et al., 2017). Yet this was not consistently the case in terms of BMI categories. An investigation of fecal microbiome in a large Chinese cohort displayed no dissimilarity of alpha diversity among BMI groups (Gao et al., 2018). Similar results were also obtained in two studies that assessed the upper digestive tract microbiome (Lin et al., 2015; Angelakis et al., 2015). Neither of these studies revealed an association between microbial diversity and BMI. These inconsistent findings with respect to bacterial diversity and its association with BMI, might not only be due to the small sample size used in our study, but also other parameters such as age, gender, and dietary consumption. Despite the above-mentioned bacterial richness estimators, OTU-level alpha diversity calculations by Shannon and Simpson indices yielded no significant difference in gut microbe diversity and richness, indicating low among-group (BMIs and T2DM) dissimilarities, which may suggest that a change of gut microbial composition might be affected by BMI or T2DM at the low taxonomic level.

Similar to previous studies (Duncan et al., 2008; Yun et al., 2017; Peters et al., 2018), there was no significant difference of the Firmicutes:Bacteroidetes ratio between BMI groups. Previous studies have yielded contradictory results with regard to a link between the ratio of Firmicutes:Bacteroidetes and obesity (Gomes, Hoffmann & Mota, 2018; Tseng & Wu, 2019). For example, higher Firmicutes:Bacteroidetes ratios were found in obese compared with non-obese Japanese subjects, and both in overweight and obese Ukrainians, respectively (Kasai et al., 2015; Koliada et al., 2017). In contrast, Schwiertz et al. (2010) found a lower proportion of Firmicutes compared to Bacteroidetes in overweight and obese volunteers. Besides these two predominant phyla, Prevotella and Bacteroides, were enriched in OB and T2DM, and in OV, respectively Enrichment of these taxa indicates enterotypes and these are likely the result of individual dietary characteristics. The presence of enterotypes in Thais has been previously shown and attributed to different diet types like vegetarians and non-vegetarians (Ruengsomwong et al., 2016). With regard to dominant gut microbiota variations, such as Firmicutes (high-fiber and carbohydrate foods) and Bacteroidetes (high-calorie foods, such as animal proteins and foods rich in fats), inconsistent results across studies could be explained by a variety of dietary components (Western or Asian diet) that may influence dynamics of gut microbiome. Therefore, detailed dietary data should be included in future research for a more comprehensive understanding of the links between dietary patterns and gut microbiota profiles.

Although we found similar profiles of the major gut bacterial phyla across the four groups, this was not the case in all taxonomic ranks. For example, F. prausnitzii, butyrate-producing bacteria, was more abundant in the OB subjects as compared to T2DM. Alteration of the gut microbiome marked by an increase of F. prausnitzii in obese or a decrease of this bacterial species in T2DM subjects has been demonstrated in several studies. Our finding is in parallel to an increase of F. prausnitzii in obese Indian and Mexican children (Balamurugan et al., 2010; Murugesan et al., 2015). A lower prevalence of this bacterium was also observed in T2DM Chinese patients in comparison with that of non-diabetic subjects (Qin et al., 2012). High accumulation of this butyrate producer in OB group may reflect the energy expenditure of the host with regard to the mechanism of its major metabolite (butyrate) in engaging host metabolism. The proof of concept for such interaction has been immensely demonstrated in an animal model (De Vadder et al., 2014; Den Besten et al., 2015) as well as in humans (Turnbaugh et al., 2006; Den Besten et al., 2013). Although the pros and cons of butyrate towards obesity has been reviewed, the capability of butyrate in influencing lipid biosynthesis could contribute to obesity (Liu et al., 2018). Collectively, these evidence highlight that the butyrate-producing species might be an indicator of host physiology.

Trends in associations between gut microbiota with some food groups and blood parameters were observed in the dominant taxa, particularly with members of Firmicutes and Bacteroidetes phyla. Prevotella and (Prevotella) were correlated with carbohydrate-rich and fiber-rich diets in OB and T2DM subjects, while their links with plant-based foods have been previously described (Kovatcheva-Datchary et al., 2015; Ruengsomwong et al., 2016; Kisuse et al., 2018). Moreover, Prevotella enterotype is generally dominant in Asian countries, where traditional high fiber diets are preferable, in contrast to Western countries, where food consumption is more homogenous (except Mediterranean diet) and mainly relies on high fat and protein content (Senghor et al., 2018). The differences in gut microbiota have been previously reported in representative Indian and Chinese subjects, as well as, Japanese populations. Variability in diets across Asia along with its geographically unique pattern contributes substantially to differences in the composition of gut bacteria communities observed in diverse Asian populations and/or ethnicities (Senghor et al., 2018; Jain, Li & Chen, 2018; Pareek et al., 2019). Notably, consumption of some types of foods (chicken, rice vermicelli, and fermented fruits or vegetables) was considerably lower in T2DM subjects than in any of the BMI groups. This is consistent with diabetic subjects being more concerned about food consumption (high cholesterol, carbohydrate, and sodium) and increased risk of complications of diabetes (Yannakoulia, 2006; Valensi & Picard, 2011; Provenzano et al., 2014; Sami et al., 2017). Although Bacteroidetes is well known to be associated with protein/fat diets (Wu et al., 2011), a positive correlation with brown rice was found in OB subjects. Whilst we did not specifically identify the species, there is evidence suggesting that an increase in abundance of some Bacteroides strains results from competition with other members for fiber-derived nutrients (Hindson, 2019). With respect to small numbers of subjects used in our study, the positive relationship of abundances of butyrate-producing bacteria (Roseburia and Faecalibacterium) with fasting glucose levels of different groups (L, OV and T2DM) may not sufficiently explain the association between these bacteria and the presence of glucose. The low abundance of these genera in T2DM patients has been previously reported (Qin et al., 2012; Karlsson et al., 2013).

Recent studies based on sequencing technology have examined the impact of gender and BMI on the status of gut microbiota (Dominianni et al., 2015; Haro et al., 2016; Borgo et al., 2018) including an aging-related decrease in gut bacteria (Shen et al., 2018; Takagi et al., 2019; Xu, Zhu & Qiu, 2019). However, our study is limited by gender-disproportionate recruitment, the age range of subjects, the variations of BMI in T2DM groups as well as, the sample size. Thus, the observed variations among different BMI groups and T2DM subjects were apparently influenced by several variables associated with female subjects. According to the MFA factor map for individuals, inclusion of confounding variables (gender, age and diets) together with blood profiles and the relative abundance of gut microbiota showed a contrary pattern between OB and T2DM. The consumption of fermented fruits or vegetables, chicken, dairy products along with the relative abundance of Faecalibacterium prevailed among OB females (made up of three women with the average age of 34), whereas a diminishing trend was displayed in older T2DM females (made up of six women with the average age of 60) with a high fasting glucose level. The worldwide trend in diabetes prevalence, including Thailand, has increased in adults over the age of 50 (Wild et al., 2004; Aekplakorn et al., 2018). Based on the average age of these two groups, OB and T2DM females seemed to differ considerably in terms of age-related changes in hormone levels. Some evidence has shown that the alteration of gut microbiota composition may be driven by the postmenopausal loss of estrogen (Vieira et al., 2017). In this study, all T2DM female subjects are postmenopausal, which the condition naturally relies on age(Agostini et al., 2018), our study suggests that a menstrual condition according to the age of the subjects should be further considered when investigating the gut microbiota profiles. In addition to other concerned factors including blood profiles, diets, BMI levels, and T2DM, the study itself may not support such a clear conclusion on the difference in the relative abundance of Faecalibacterium derived from the age differences in subjects. A possible aspect that could be drawn from such differences marked in OB and T2DM may be the specificity of metabolic diseases in gut bacteria associations (Festi et al., 2014; Gurung et al., 2020). Considering all blood profiles in association with fecal gut microbiota (irrespective of gender and age), neither the blood profiles nor the gut microbiome influenced on a specific group of subjects. Besides, the individual differences shown on the MFA were resulted from some bacterial genera (Bacteroides, Prevotella and Faecalibacterium) and blood profiles (total cholesterol, triglyceride, and diastolic blood pressure, systolic blood pressure and HDL cholesterol) that only contributed to the variation of subjects. Evidence for the involvement of gut microbiome in metabolic disorders that posed either a detriment (Dabke, Hendrick & Devkota, 2019; Gildner, 2020) or a benefit to host health (He & Shi, 2017) encourage further study to increase the sample size as our conclusion might not be applicable for the study with a large population. Therefore, this adjustment may help to facilitate explaining the associations between blood profiles and the gut bacteria.

Conclusions

Our study has provided a preliminary overview of prokaryotic communities in the gut of adult Thais, regardless of the small sample size. Associations between dietary intake, blood profiles, and fecal gut microbiome in different BMI and T2DM subjects were also examined. A range of multivariate data analysis (MFA and sPLS) enabled us to capture the profiles of individuals in each study group. Subjects with obesity and/or diabetes might be associated with different bacterial populations when linking with dietary consumption and blood profiles. However, a larger sample size is mandatory to advance an understanding of the interplay of BMI or T2DM to changes of microbiota composition, together with metabolomics data. Validation of abundance of considered taxa related to BMIs by qPCR should be additionally included in future research.

Supplemental Information

Supplemental Information 1 The frequency of dietary consumption of subjects in BMI and T2DM groups.

The statistical significance of differences in the mean ranks among groups was determined using Kruskal-Wallis rank sum test with Benjamini–Hochberg procedure (p value adjusted < 0.05).

Click here for additional data file.

Supplemental Information 2 Sequencing data processing of each sample.

Click here for additional data file.

Supplemental Information 3 OTU richness and diversity indices of each sample.

Click here for additional data file.

Supplemental Information 4 MFA analysis of frequency of food consumption profiles in different BMI groups and T2DM subjects.

The factor map presents the frequency of food consumption profiles in different BMI groups and T2DM subjects based on MFA. The coordinates of the individuals are indicated by the 95% confidence ellipses including orange (Lean), blue (OV), green (OB), and purple (T2DM).

Click here for additional data file.

Supplemental Information 5 Rarefaction curves of fecal microbial diversity estimators of 30 subjects.

Sampling curves generated for each fecal sample show the relationship between a diversity index (vertical axis) and sequencing depth (horizontal axis). (A) Rarefaction curve based on Chao1. (B) Rarefaction curve based observed species. (C) Rarefaction curve based Shannon. Each curve is represented by a unique color.

Click here for additional data file.

Supplemental Information 6 Fecal microbial profiles in BMI groups and T2DM group.

(A) Relative abundance of fecal gut microbiome at phylum level presented in each subject. (B) Relative abundance of fecal gut microbiome at phylum level presented in each group. (C) OTU annotation tree of bacterial genera. (D) Boxplot of Firmicutes to Bacteroidetes ratio across four groups (asterisks indicate p < 0.05, Wilcoxon rank-sum test).

Click here for additional data file.

Supplemental Information 7 Relative abundance of dominant species in BMI groups and T2DM group.

(A) Relative abundance of fecal gut microbiome at species level presented in each subject. (B) Relative abundance of fecal gut microbiome at species level presented in each group. (C) Boxplot of relative abundance of Faecalibacterium prausnitzii across four groups. (D) Boxplot of relative abundance of Prevotella copri across four groups. Asterisks indicate p < 0.05, Wilcoxon rank-sum test without Benjamini–Hochberg method.

Click here for additional data file.

Supplemental Information 8 Beta diversity analysis at phylum level.

Differences of fecal bacterial community structure between BMI and T2DM groups evaluated by beta diversity analysis. (A) UPGMA cluster tree based on Weighted UniFrac distance. (B) UPGMA cluster tree based on Unweighted UniFrac distance (below). Subjects from L, OV, OB, and T2DM groups are presented in blue, green, red, and yellow color, respectively.

Click here for additional data file.

Supplemental Information 9 Associations between gender, age, blood profiles (HDL and FBS), dietary habits, and fecal gut microbiota in BMI and T2DM groups.

The factor map presents the integration of gender, age, blood profiles (HDL and FBS), dietary habits, and fecal gut microbiota (at genus level) of subjects in different BMI groups and T2DM group based on the MFA. F: a female individual, M: a male individual.

Click here for additional data file.

Supplemental Information 10 Associations between gender, age, all blood profiles, dietary habits, and fecal gut microbiota in BMI and T2DM groups.

The factor map presents the integration of gender, age, all blood profiles, dietary habits, and fecal gut microbiota (at genus level) of subjects in different BMI groups and T2DM group based on the MFA. F: a female individual, M: a male individual.

Click here for additional data file.

Supplemental Information 11 Associations between all blood profiles and fecal gut microbiota in BMI and T2DM groups.

The factor map presents the integration of all blood profiles and fecal gut microbiota (at genus level) of subjects in different BMI groups and T2DM group based on the MFA. The coordinates of the individuals are indicated by the 95% confidence ellipses including green (L), orange (OV), purple (OB), and pink (T2DM).

Click here for additional data file.

Supplemental Information 12 Questionnaire of participant characteristics and food frequency.

Click here for additional data file.

Supplemental Information 13 Questionnaire responses on characteristics and frequency of dietary consumption of each subject.

Raw questionnaire data collected from subjects that participated in this study.

Click here for additional data file.

The authors would like to thank all volunteers for providing dietary consumption data, blood and fecal samples. We would like to thank Mr. Tawatchai Chumponsuk for technical assistance. We would like to thank Associate Professor Jiro Nakayama for useful comments. We thank Dr. Eleni Gentekaki for useful discussions and editing of this manuscript.

Additional Information and Declarations

Competing Interests

Author Contributions

Human Ethics

Data Availability

Kongkiat Kespechara is the president of Sooksatharana (Social Enterprise) Co., Ltd, Thailand.

Lucsame Gruneck conceived and designed the experiments, performed the experiments, analyzed the data, prepared figures and/or tables, authored or reviewed drafts of the paper, and approved the final draft.

Niwed Kullawong conceived and designed the experiments, performed the experiments, authored or reviewed drafts of the paper, and approved the final draft.

Kongkiat Kespechara conceived and designed the experiments, authored or reviewed drafts of the paper, and approved the final draft.

Siam Popluechai conceived and designed the experiments, performed the experiments, analyzed the data, prepared figures and/or tables, authored or reviewed drafts of the paper, and approved the final draft.

The following information was supplied relating to ethical approvals (i.e., approving body and any reference numbers):

This study was approved by the Ethics Committee of Mae Fah Luang University (ethics license: REH60075).

The following information was supplied regarding data availability:

Sequences of the 16S rRNA gene of fecal microbiota are available at BioSample: SAMN14309526–SAMN14309555 (also available at BioProject: PRJNA610672 or SRA SRP251788).

Corresponding sequences (Fastq format) are available at https://trace.ncbi.nlm.nih.gov/Traces/sra/?study=SRP251788.

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
