# Peer review of "Gut microbiota of obese and diabetic Thai subjects and interplay with dietary habits and blood profiles"

_PeerJ, doi:10.7717/peerj.9622_

## Round 0.1 · original submission · Major Revisions

Although the reviewers and myself have seen interesting findings in your work they have raised several concerns and questions that need to be addressed. It is particularly important to focus more on the introduction (some suggestions to improve clarity have been indicated), and also smooth the message of your conclusions due to the small numbers of individuals you have sampled. Also, you need to take into account other factors and characteristics of the population sampled that may have an influence on the results (age, sex, etc.).

Reviewer 1 ·

Basic reporting

The manuscript is presented in an intelligible fashion and written in standard English. The article include sufficient introduction and background to demonstrate how the work fits into the broader field of knowledge. However, there are several aspects of format that should be improved before publication:

1) The scientific name of the taxa at any taxonomic level must appear on italic, for example, line 250-251 and line 295.
2) In Table 2, in which the sPLS analysis is shown, the authors should please specify in some way, which associations correspond to sub-index a and which to sub-index b, since the relevance is not the same.
3) The authors reveal that they have observed significant differences were noted regarding consumption of chicken (0.005), rice vermicelli (0.028) and fermented fruits or vegetables (0.038) between groups, but could the authors be clearer and to specify between which groups in the text (Line 201)?
4) Line 263: Faecalibacterium showed significant difference between OV-OB and OB-T2DM comparisons (Fig. 3C), while [Prevotella] differed in L and OV comparison (Fig. 3D). What is the P-value and the P-value adjusted? Please, there are more cases like this in the text, please include this information.
5) The clarity and resolution of figure 4 is insufficient and tends to confuse, I recommend an improvement in it for a more intuitive observation of the results. In this same figure, DiabeM refers to the T2DM group? Why was it named differently in the first graph of figure 4?
6) Please check the nomenclature of the groups of study, for example in the Line 271, groups V and B are mentioned, when you should put OV and OB, It is not like this?
7) There is an error with respect to figure 6, in the text (Line 301) it is indicated "Dim 1 and 2 with large eigenvalues (2.28 and 1.96, respectively)" while the figure 6 shows other values Dim 1 = 12.8% and Dim 2 = 14.3%.

On the other hand, the authors provide in the supplementary material, the patients data (age, gender, weight, HDL-c, fasting glucose level, etc.), sequencing data processing of each sample, OTU richness and diversity indices of each sample, the food frequency questionnaire and the results of FFQ of each patient, by contrast, the authors not mentioned data deposition statement or at least the deposition information not is noted in the manuscript. According to Peer J policy the raw data, in this case fastq files must be publicly deposited.

Finally, the article complies with the ethical requirements and with the data protection of the subjects.

Experimental design

The topic of the manuscript is interesting but there are some problem.

The aimed of this paper to explore whether microbial composition and diversity has an association with host BMI and T2DM in a population of Thai subjects. However, the sample size is very small to each group of study with 8 subject in the lean and overweight groups and 7 subjects in the obese and T2DM groups.

Furthermore, the authors have not taken into account the homogeneity of the groups in terms of gender, especially it should be noted that the obese group is made up of 4 men and 3 women and the T2DM group is made up of 6 women and a single man. On the other hand, the ages are highly variable, finding a 71-year-old individual in the obese group, the youngest of this group being a subject with 24 years, so the average age of the group is 42 years. However, the T2DM group has an average age of 60 years, which added to the fact that it is made up of women almost exclusively, we could say that they are postmenopausal women.
These gender and age aspects are especially important when dealing with the analysis of the intestinal microbiota.

To this we must add that in the T2DM group we found 4 thin, 2 overweight and one obese. Therefore, when attributing the changes in the intestinal microbiota in this group with respect to the obese group, we cannot assure that it is due solely to the fact of having diabetes. Furthermore, if we see in Table 1, the characteristics of the study subjects, both the obese group and the T2DM present 3 criteria for metabolic syndrome, hypertension, high triglyceride levels and elevated fasting glucose.

Taken together all these aspects, the authors must better define the population under study, in order to reach the objective they are addressing in this article.

Validity of the findings

The topic of the manuscript is interesting and It is also novel because the intestinal microbial community is investigated in a population of Thai subjects, not previously described.

The methodology is described in sufficient detail and the analysis of the data. The statistical analysis appears to be technically sound. However, I would appreciate a more detailed explanation about the MFA analysis, as well as its results and its importance in the manuscript.

One aspect that authors should address is the use of QIIME 2 instead of QIIME 1. QIIME has been shown not to work optimally with baseline settings, furthermore this paper from 2010 does not show default settings for MiSeq. Why is the QIIME2 version not used? Please elaborate.

The same goes for the used version of the Greengene database (is not mentioned in the manuscriopt), even if it was the latest version of it, this is years old and in the meantime a lot has changed in taxonomy especially in phylotypes associated with the human GI tract. I would suggest the latest SILVA database.

The authors also recognize the weaknesses of the study population and the need to complement with other techniques for a deeper understanding of the subject.

Perhaps the real goal of this study should be to describe the gut microbiota in relation to BMI and T2DM in Thai subjects and to evaluate possible microbial patterns associated with their eating habits and blood profiles. Please consider this on Line 69.

Additional comments

The work in this study describes a cross-sectional 16S rRNA analysis of the bacterial community structure of the gut microbiota in 30 subjects stratified into three categories based on BMI and a group with diabetes as well as the relationship with their eating habits and blood profiles in a population of Thai subjects.

The first issue stems from the fact that the sample size is very small for a microbiome study that was designed to assess the role of BMI and T2DM on the composition of the gut microbiome. Indeed, the authors themselves point out that there have been many conflicting results published on changes related with obesity and T2DM in the gut microbiota that likely reflect differences in the populations under study, small cohort sizes, complexity of the obesity phenotype, confounding factors such as age, gender, diet, etc. Other published reports have described gender-related differences in the microbiome from the gut (Mueller et al., AEM, 2006; Markle et al., Science, 2013; Santos-Marcos et al., Mol Nutr Food Res., 2019; Kim et al., World J Mens Health, 2020). In addition, In this manuscript both the obese and T2DM groups have metabolic syndrome, according to the National Cholesterol Education Program Adult Treatment Panel III (NCEP-ATPIII) the subjects have elevated fasting glucose, hypertension and elevated triglycerides levels. In this sense, the recent literature supports the gut microbiome’s potential influence on the various risk factors of metabolic syndrome (Gildner., Evol Med Public Health, 2020; Dabke et al., J Clin Invest, 2019). Finally, the group of diabetics is much older, this could be influencing the composition of the microbiota (Nagpal et al., Nutr Healthy Aging, 2018; Maynard et al., Subcell Biochem, 2018) also the majority of subjects in T2DM group are women and have been observed changes in the intestinal microbiota in post-menopausal women (Vieira et al., Front. Microbiol, 2017). All these works suggest the potential involvement of differences in gut microbiota in the unequal incidence of metabolic diseases. How might these findings influence your results? The differences between and significance of the findings from this study in light of these previously published results should be discussed. At a minimum the authors should discuss the power calculations that were used to determine the size of their cohort. If this cannot be done, then the conclusions need to be tempered to say that their conclusions are valid for this set of subjects only and may or may not be generalizable to a larger population study.

Last, antibotic usage in the analysis of the gut microbiota is also important. Have subjects taken antibiotics or prebiotics / probiotics or some type of drug prior to evaluation of the gut microbiota?

Reviewer 2 ·

Basic reporting

Grammar and punctuation corrections needed

Experimental design

1. Are there any sampling bios? If so how to rectify that.
2. selection criteria for the subject are required.

Validity of the findings

Faecalibacterium prausnitzii, a butyrate-producing bacterium, significantly higher in OB than that in other groups-discuss about this

Reviewer 3 ·

Basic reporting

This manuscript is an observational study of Thai adults. Subjects were classified according to their obesity and diabetes, to compare the differences between their eating habits and gut microbiota.

1) The language is not bad.
2) To clarify the purpose of the study, it is necessary to modify the paragraph structure of the introduction.

1st: An overview that obesity and diabetes are continuously increasing in recent years and are becoming a socioeconomic burden.
2nd: Overview of the growing interest in research on intestinal microbiota and correlation with various diseases
3rd: Explain that obesity is universally described as BMI, along with existing studies showing that westernized eating habits are related to some diseases
4th: Explaining that there is a need to check the relationship between eating habits and gut microbiota depending on the obesity and diabetes of Thai citizens.

3) Nevertheless, this study seems to have one major flaw in the hypothesis. This study compared an average of 60's obese/overweight diabetics with the general population, and it is difficult to say that these subjects represent diabetics.

Experimental design

In the extension of 3) mentioned above, I'd like to add more details.

Subjects were divided into 3 groups according to the obesity level.
How about comparing the difference between diabetes and non-diabetes in each group after diabetics are included in the group according to obesity level?

If the baseline characteristics start with the existence of differences between groups, the meaning of comparison seems to be diminished.

Validity of the findings

Gut micriobiome(OR microbiota) studies in populations that do not focus on westernized diets have been conducted in several countries. In particular, I'd like to recommend that the authors search for additional studies targeting populations in East Asia.

Additional comments

No comment

---

## Round 0.2 · accepted · Accept

Thanks for addressing all points and concerns raised by the two reviewers. I believe your manuscript has greatly improved and now is ready for publication on PeerJ.

Reviewer 3 ·

Basic reporting

Manuscripts were written in an easy-to-understand manner and in standard English.

The composition of the paragraphs pointed out in the INTRODUCTION section has been revised as appropriate to reflect the reviewer's points.

Experimental design

The sample size is still very small, which will remain the biggest weakness of this paper.
No matter how gender and age are considered, it is difficult to say that these analysis results represent typical Thai adults.

Validity of the findings

Despite the methodological limitation, the results described in this paper are interesting. This is because the authors have observed changes in the gut microbiota of Thai adults that have not been previously identified.

Additional comments

No comment.